# Spatial distribution and determinants of micronutrient intake status among children aged 6–23 months in Ethiopia: A Multi-scale Geographical Weighted Regression Analysis

**Alemu Birara Zemariam**[1]*, **Addis Wondmagegn Alamaw**[2], **Rediet Woldesenbet Molla**[3], **Tesfaye Engdaw Habtie**[4], **Molla Azmeraw Bizuayehu**[1], **Ribka Nigatu Haile**[1], **Tegene Atamenta Kitaw**[4], **Biruk Beletew Abate**[1], **Mollalign Aligaz Adisu**[1]

1 Department of Pediatrics and Child Health Nursing, School of Nursing, College of Medicine and Health Science, Woldia University, Woldia, Ethiopia, 2 Department of Emergency and Critical Care Nursing, School of Nursing, College of Medicine and Health Science, Woldia University, Woldia, Ethiopia, 3 Department of Midwifery, School of Midwifery, School of Midwifery, College of Medicine and Health Science, Woldia University, Woldia, Ethiopia, 4 Department of Nursing, School of Nursing, College of Medicine and Health Science, Woldia University, Woldia, Ethiopia.

* alexb7298@gmail.com

## Abstract

### Introduction

Micronutrient (MN) deficiency is a major global health concern, especially in developing countries like Ethiopia. However, there is a lack of comprehensive monitoring and information regarding MN intake status. This study aimed to assess the spatial distribution and factors influencing the intake of MN-rich foods among children aged 6–23 months in Ethiopia.

### Methods

This study utilized the Ethiopian 2016 Demographic and Health Survey dataset with 2562 weighted children. The Bernoulli model was applied using Kuldorff's SaTScan version 9.6 software. The spatial distributions of MN-rich food intake were visualized using ArcGIS pro version 3.0 software. Model comparison was conducted using log likelihood ratio and corrected Akakie Information Criteria. A multi-scale geographical weighted regression analysis was performed using MGWR version 2.0 software. A P-value threshold of less than 0.05 was used to identify spatially significant predictors.

### Results

Overall, 69% (95% CI: 60.87, 77.43) of children aged 6–23 months in Ethiopia were found to have consumed foods rich in MN. The intake of these nutrient-rich foods exhibited significant clustering in specific regions, including Addis Ababa, Dire Dawa,

**Data availability statement:** The original contributions presented in the study are included in the article; further inquiries can be directed to the corresponding author via alexb7298@gmail.com.

**Funding:** The author(s) received no specific funding for this work.

**Competing interests:** The authors have declared that no competing interests exist.

**Abbreviations:** AICc-corrected akakie information criteria, ANC-antenatal care, GWR-geographic weighted regression, MGWR-multi-scale geographic weighted regression, MNP-multiple micronutrient powder, OLS-ordinary least square, MN-micronutrient, EDHS-Ethiopian demographic and health survey, VA-vitamin A

Harari, certain parts of Benishangul, and the Gambella region. Spatial scan statistics analysis identified a total of 65 primary spatial clusters. Children residing within the primary clusters were found to be 29% more likely to have an intake of foods rich in MN compared to those living outside the identified clusters (RR = 1.29, LLR = 25.34, P < 0.001). Key spatially significant predictors included higher household wealth status; children aged 13–23 months, the presence of antenatal care, and mothers with any job.

## Conclusion

The consumption of foods rich in MN in Ethiopia displayed non-random spatial patterns. To tackle the problem of inadequate intake of these nutritious foods and address the burden of MN deficiency among children aged 6–23 months, it is essential for policymakers and health planners to implement targeted nutrition interventions in the identified areas and factors.

## Introduction

Malnutrition is the condition that develops when the body is deprived of vitamins, minerals and other nutrients it needs to maintain healthy tissues and organ function. Malnutrition occurs in people who are either undernourished or overnourished [1]. Micronutrient (MN) deficiency is the global public health concern among children and it is worse in developing countries including Ethiopia [2–4]. Notably, MN during the first 1000 days of child life is critical for optimal growth, health and development, but little attention has been given to MN supplementation status [5]. Children, especially those in low and middle-income countries like Ethiopia, are particularly vulnerable to under nutrition [2].

Globally, malnutrition remains a significant issue, with one in three children under the age of five not receiving sufficient nutrition. MN deficiencies often go unnoticed, leading to around 340 million children worldwide experiencing what is termed "hidden hunger"[6]. Children need to receive adequate nutrient supplementation to support their growth and overall well-being. MN is basically classified into two groups like vitamins and minerals. Iron, zinc, iodine, vitamins like vitamin A and folic acid (vitamin B9) are the listed micro-nutrients associated with the cause of mortality and morbidity in children aged under 5 [7]. Despite being needed in smaller quantities, MN contribute significantly to the production of hormones, enzymes, and other essential substances necessary for growth and well-being [8].

In Ethiopia around 37.3% children aged 6–23 months does not get enough MN supplement according to their age [9]. Reports from national surveys indicate concerning levels of MN deficiencies among Ethiopian children. According to the 2016 Ethiopian Demographic and Health Survey (EDHS), approximately 38% of children aged 6–59 months in urban areas and 24% in rural areas suffer from iron or folate deficiency. Iron deficiency, in particular, is a major cause of anemia, affecting 54% of children in this age group. Alarmingly, only 9.24% of these children receive

iron supplementation [10,11]. Additionally, vitamin A deficiency alone is responsible for 80,000 deaths among preschool children in Ethiopia, with only 63% of children nationwide receiving adequate supplementation which is far from the government plan of 90%. Furthermore, approximately 35% of Ethiopian children suffer from zinc deficiency, despite its crucial role in immune system function and overall well-being [12,13].

Various nations employ different strategies to address MN deficiency and enhance their intake in children, including supplementation, fortification, food diversification, and public health interventions. The World Health Organization recommends MN supplementation for children due to the significant impact of deficiency on their well-being, providing guidelines to support these efforts [7,14]. Studies support the effectiveness of supplementation programs in addressing MN deficiency, either through individual supplementation of specific micronutrients like zinc, iodine, iron, or vitamin A, or through multiple MN supplementation combining three or more micronutrients in a single dose [15].

Parental educational status, maternal occupational status, antenatal care (ANC) follow up of the mother, working status of the caregiver, birth weight of the child, wealth index, residency area, geographical differences of the country are listed among the factors which are associated with MN intake [9,16]. Micronutrient deficiencies are widespread among Ethiopian children aged 6–23 months, with inadequate intake of essential vitamins and minerals leading to health issues like impaired cognitive development and increased infection risk. Geographic disparities in nutrient availability and dietary practices highlight the impact of socio-economic factors and local food systems. Despite existing research, there is a lack of understanding regarding the spatial distribution of micronutrient intake since most studies focus on local or small settings, neglecting multi-scale geographical variations and national level determinants of nutrient intake, hindering the development of geographically targeted interventions. Furthermore, data on hot and cold spot areas for MN intake is scarce. To expedite progress towards improving child health and ending child malnutrition, understanding the geospatial distribution of MN intake and its determinants among children is crucial. Thus, this study aims to better understand the spatial pattern of MN intake status and its determinants among children aged 6–23 months in Ethiopia using the nationally representative data.

## Methods

### Study setting, data source, and study period

We utilized the 2016 EDHS, which was a cross-sectional data in nine geographical region namely Tigray, Afar, Amhara, Oromia, Somalia, Benishangul-Gumuz, Southern Nation Nationality, and People's Region (SNNPR), Gambella, and Harari and two administrative cities (Addis Ababa and Dire Dawa).

We obtained permission and ethical approval from DHS data archivist to extract the data from www.measuredhs.com. The EDHS surveys are carried out roughly every 5 years aimed at generating updated health and health-related indicators. Sample weights were applied to ensure the representativeness of the samples and account for the complex survey design, survey non-response, and post-stratification. A multistage stratified sampling technique was used. The study focused on children aged 6–23 months using the kid's records (KR) file. After performing data cleaning and exploration, our analysis included a total weighted sample of 2562 children aged 6–23 months with a total of 21 features [17].

### Source and study population

The target population for this study consisted of all living children aged 6–23 months with their mothers in Ethiopia whereas, all children aged 6–23 months in the selected enumeration area were the study population.

### Variables and measurements

The outcome of interest was the status of micronutrient intake. **Micronutrient intake** was assessed by taking into account different elements, including the consumption of foods rich in Vitamin A or iron within the last 24 hours, the use of multiple

micronutrient powder (MNP) or iron supplements in the preceding 7 days, and the administration of vitamin A supplements or deworming treatment within the past 6 months [18]. Consequently, if the respondent reported that the child had eaten' at least one of the minimum recommended MNs, we considered it "Yes"; if the children received none of the minimum recommended MNs, it was considered as "No".

To assess the consumption of foods rich in vitamin A (VA), we analyzed the intake of seven specific food groups within the previous 24 hours. These food groups included eggs, various meats (such as beef, pork, lamb, and chicken), pumpkin, carrots, and squash, dark green leafy vegetables, mangoes, papayas, and other fruits rich in VA, as well as liver, heart, and other organs, and fish or shellfish. Similarly, we assessed the consumption of iron-rich foods by examining the intake of four specific food groups within the previous 24 hours. These food groups consisted of eggs, various meats, liver, heart, and other organs, as well as fish or shellfish. To determine the intake of MNP, we asked the respondents if their child had received such powders in the past seven days. For assessing iron supplementation, we inquired whether the child had been given iron pills, sprinkles with iron, or iron syrup within the past seven days. The researchers examined vitamin A supplementation (VAS) and deworming treatment by reviewing the integrated child health card, which contains information on immunization and growth monitoring history. They also obtained verbal responses from the mothers. These assessments were specifically conducted for children aged 6–23 months to determine if they had received VAS and deworming treatment in the last six months. If the respondent reported that the child had consumed at least one of these supplementations, it was categorized as a "Yes" response, indicating the consumption of MN-rich foods.

This study considered different independent variables to identify determinants of MN intake among children aged 6–23 months in Ethiopia (Table 1).

## Data processing and analysis

The data was obtained from the EDHS 2016 dataset, and STATA version 17 was employed for descriptive statistics. The data underwent preprocesses steps including weighting, cleaning, and recoding. For mapping the MN intake status at

**Table 1. List of independent variables for the assessment of spatial distribution of micronutrient intake and its predictors among children aged 6-23 months in Ethiopia using 2016 EDHS.**

| Variable | Descriptions (classification) |
| --- | --- |
| Sex of the child | Male or female |
| Age of the child | 6-12 month, 13–23 month |
| Mother age | 15–24, 25–34, above 35 years |
| Residence status | Urban or Rural |
| Region | Tigray, Amhara, Oromia, SNNPR, Benishangul, Afar, Gambela, Somali, Harari, Addis Ababa, and Dire Dawa |
| Mother educational status | No education, Primary, Secondary, and Higher |
| Mothers working status | Working or not working |
| Wealth index | Poor, Middle, and Rich |
| Desire more children | No more want, undecided, wants more |
| Antenatal follow-up | No or Yes |
| Place of delivery | Home or health institution |
| Postnatal care | No or Yes |
| Sex of household head | Male or Female |
| Media access/ exposure | Yes or No |
| Child had cough in the last 2 weeks | Yes or No |
| Family size | < 5 and 5 or above |
| Child had diarrhea in the last 2 weeks before the survey | Yes or No |

regional levels and identify spatially significant predictors, ArcGIS Pro version 3.0, Sat Scan version 9.6, and MGWR version 2.0 were employed.

## Spatial autocorrelation and hot spot analysis

The Global Moran's I statistic was used to evaluate the regional variation of micronutrient intake among children aged 6–23 months in Ethiopia. When the Moran's I values are close to -1, it indicates that the consumption of micronutrient is poor and dispersed. On the other hand, values close to + 1 indicate that the consumption is good and clustered. A Moran's I value of zero suggests that the consumption is randomly distributed across regions. Spatial autocorrelation is declared at a statistically significant Moran's I p-value less than 0.05 [19]. Additionally, there is a type of autocorrelation known as incremental autocorrelation that occurs at different distances. Z-scores for spatial autocorrelation were calculated at various distances to assess the strength and statistical significance of spatial clustering. The highest Z-scores indicate the distances at which spatial grouping is most pronounced [20].

Additionally, the Hot Spot Analysis was conducted using Getis-Ord Gi* Spatial Statistics in order to identify spatial clusters that are statistically significant in terms of having either low (cold spots) or high (hot spots) levels of micronutrient intake. Clusters with high Gi* values indicate significant hot spots, whereas clusters with low Gi* values indicate significant cold spots.

## Spatial interpolation

Ordinary Kriging interpolation technique was utilized to predict the MN intake status among children aged 6–23 months in the unsampled enumeration areas based on the tested population. This approach relaxes the assumption of a Gaussian distribution of the observed semi-variogram in the input data, which is often not realistic in practice [21]. This interpolation method generates a new simulated semi-variogram at each location, taking into account the estimated semi-variogram from the input data. The weight of the new simulated semi-variogram is calculated using Bayes' rule [22].

## Spatial scan statistics

We utilized a spatial scan statistics model based on Bernoulli distribution to identify the geographic areas where there are statistically significant clusters of micronutrient intake among children aged 6–23 months. This analysis was conducted using Kuldorff's SaTScan version 9.6 software [23]. A moving scanning window was employed to assess the distribution of micronutrient intake among children aged 6–23 months in the study area. The scanning window was circular in shape and the identification of the most likely clusters was determined using P-values and log-likelihood ratio tests based on 999 Monte Carlo replications.

## Model evaluation

### Spatial regression analysis

**Ordinary least squares (OLS) regression.** Once the hot spot areas were identified, spatial regression modeling was conducted to explore the factors influencing the observed spatial clustering. Initially, OLS regression analysis was fitted. The OLS model is a global model that estimates a single coefficient for each explanatory variable across the entire study area. Global models assume that the factors affecting micronutrient intake are consistent and do not vary geographically. However, this assumption of geographical independence may introduce bias into the parameter estimates. To assess the assumption of spatial dependency, the Koenker (BP) statistics was used which were statistically significant (P < 0.001), indicated that the relationships were not consistent, possibly due to non-stationary or heteroscedasticity.

In the presence of spatial dependency, the coefficient of the independent variable varies locally, meaning that the predictor variables may or may not be significant in different local areas [24]. Therefore, it suggested further regression

model to account their huge limitations such as geographically weighted regression (GWR) and multiscale geographically weighted regression (MGWR).

**Geographically weighted regression.** The variation in the relationship between variables across might not be the same across different clusters and it can be identified using GWR. Unlike OLS regression, which fits a single linear regression equation to all the data in the study area, GWR creates a separate equation for each cluster in the DHS data. OLS regression calibrates the equation using data from all features or clusters, whereas GWR uses data from nearby features or clusters. As a result, the coefficient in GWR takes on different values for each cluster [25]. To explore the varying relationships between micronutrient intake and explanatory variables in different geographic areas, a newly emerging geographic regression model called MGWR was employed. Unlike the traditional GWR model, MGWR not only allows coefficients to vary across space but also accommodates variations in scale among different covariates. MGWR takes into account different spatial scales by considering distinct neighborhoods for each explanatory variable, thereby capturing the complex spatial relationships between variables [26]. To assess the presence of multicollinearity between variables, the Variance Inflation Factor (VIF) was calculated. A VIF value exceeding 7.5 indicates the existence of multicollinearity [27]. In order to select the most appropriate model, the Corrected Akakie Information Criterion (AICc) and adjusted R2 values were computed. The model with the lowest AICc and the highest adjusted R2 was considered the best-fitting model [28]. In all statistical analyses, a P-value threshold of less than 0.05 was used to determine statistical significance.

**Ethics approval and consent to participate.** The 2016 EDHS was ethically reviewed by the National Research Ethics Review Committee of the Ethiopian Ministry of Science and Technology. As described in the survey final report, involvement in the survey program was voluntary, and verbal agreement (informed consent) was also taken [29]. The International Review Board of Demographic and Health Surveys program data archivists were allowed to download and use the datasets for this study. Also, the data was handled according to the Helsinki Declaration of the World Medical Association.

## Results

### Socio-demographic characteristics of the study participants

A total of 2562 children aged 6–23 months were included in this study. The mean ± SD age of the children was 11.92 ± 4.27 months. More than half (52.3%) of the child mothers were in the age group of 25–34 years. Nearly half (45.0%) of the households had poor wealth status and majority (75.5%) of the respondents were reside in rural area. Regarding maternal health service utilization, more than two-third (75.1%) of respondents were had ANC follow-up and nearly half (48.5%) of the respondents were gave birth at health facility. More than one-third (34.5%) of the female children had consumed foods rich in micronutrients. Of the total, 1062 (41.5%) of children whose age 13–23 months had taken foods rich in micronutrient (Table 2).

### Spatial distribution of micronutrient intake status among children aged 6–23 months

The prevalence of adequate micronutrient intake among children aged 6–23 months was 69.01% (95% CI: 60.87%, 77.43%). The regions with the highest rates of micronutrient intake among children aged 6–23 months were Tigray, SNNPR, Oromia, Gambella, and Benishangul Gumuz (Fig 1).

### Spatial and incremental autocorrelation analysis

The spatial distribution of micronutrient intake among children aged 6–23 months in Ethiopia is clustered with Global Moran's I value of 0.167, p-value <0.001, and Z-score of 6.17. Thus, micronutrient supplementation status has a spatial dependency. In addition, the likelihood of this clustered pattern because of random chance is less than 1% (Fig 2).

**Table 2. Socio-demographic characteristics of study participants 2016 EDHS.**

| Variables | Categories | Micronutrient intake status | | | | Total Weighted (n=2562) | |
|---|---|---|---|---|---|---|---|
| | | No | | Yes | | | |
| | | Count | Percent | Count | Percent | Count | Percent |
| Respondent's current age | 15-24 | 242 | 9.4% | 520 | 20.3% | 762 | 29.4% |
| | 25-34 | 390 | 15.2% | 925 | 36.1% | 1315 | 52.3% |
| | >=35 | 162 | 6.3% | 323 | 12.6% | 485 | 18.3% |
| Region | Tigray | 51 | 2.0% | 227 | 8.9% | 278 | 12.8% |
| | Afar | 120 | 4.7% | 124 | 4.8% | 244 | 7.0% |
| | Amhara | 99 | 3.9% | 130 | 5.1% | 229 | 7.4% |
| | Oromia | 126 | 4.9% | 238 | 9.3% | 364 | 13.5% |
| | Somali | 151 | 5.9% | 166 | 6.5% | 317 | 9.4% |
| | Benishangul | 29 | 1.1% | 176 | 6.9% | 205 | 10.0% |
| | SNNPR | 86 | 3.4% | 241 | 9.4% | 327 | 13.6% |
| | Gambella | 40 | 1.6% | 127 | 5.0% | 167 | 7.2% |
| | Harari | 42 | 1.6% | 114 | 4.4% | 156 | 6.4% |
| | Addis Ababa | 26 | 1.0% | 108 | 4.2% | 134 | 6.1% |
| | Dire Dawa | 24 | 0.9% | 117 | 4.6% | 141 | 6.6% |
| Residence | Urban | 98 | 3.8% | 434 | 16.9% | 532 | 24.5% |
| | Rural | 696 | 27.2% | 1334 | 52.1% | 2030 | 75.5% |
| Respondent's highest educational level | No education | 568 | 22.2% | 955 | 37.3% | 1523 | 54.0% |
| | Primary | 168 | 6.6% | 545 | 21.3% | 713 | 30.8% |
| | Secondary | 36 | 1.4% | 171 | 6.7% | 207 | 9.7% |
| | Higher | 22 | 0.9% | 97 | 3.8% | 119 | 5.5% |
| Religion | Orthodox | 189 | 7.4% | 571 | 22.3% | 760 | 32.3% |
| | Muslim | 470 | 18.3% | 809 | 31.6% | 1279 | 45.8% |
| | Protestant | 120 | 4.7% | 340 | 13.3% | 460 | 19.2% |
| | Others** | 15 | 0.6% | 48 | 1.9% | 63 | 2.7% |
| Family size | <5 | 260 | 10.1% | 583 | 22.8% | 843 | 33.0% |
| | >=5 | 534 | 20.8% | 1185 | 46.3% | 1719 | 67.0% |
| Sex of household head | Male | 622 | 24.3% | 1454 | 56.8% | 2076 | 82.2% |
| | Female | 172 | 6.7% | 314 | 12.3% | 486 | 17.8% |
| Wealth index combined | Poor | 503 | 19.6% | 795 | 31.0% | 1298 | 45.0% |
| | Medium | 92 | 3.6% | 284 | 11.1% | 376 | 16.1% |
| | Rich | 199 | 7.8% | 689 | 26.9% | 888 | 39.0% |
| Currently pregnant | No or unsure | 737 | 28.8% | 1655 | 64.6% | 2392 | 93.6% |
| | Yes | 57 | 2.2% | 113 | 4.4% | 170 | 6.4% |
| Desire for more children | Wants more | 567 | 22.1% | 1214 | 47.4% | 1781 | 68.7% |
| | Undecided | 34 | 1.3% | 80 | 3.1% | 114 | 4.5% |
| | Wants no more | 193 | 7.5% | 474 | 18.5% | 667 | 26.8% |
| Father's education level | No education | 446 | 17.4% | 691 | 27.0% | 1137 | 39.1% |
| | Primary | 220 | 8.6% | 642 | 25.1% | 862 | 36.3% |
| | Secondary | 70 | 2.7% | 241 | 9.4% | 311 | 13.6% |
| | Higher | 58 | 2.3% | 194 | 7.6% | 252 | 11.0% |
| Father's occupation | Not working | 153 | 6.0% | 304 | 11.9% | 457 | 17.2% |
| | Working | 641 | 25.0% | 1464 | 57.1% | 2105 | 82.8% |
| Respondent's occupation | Not working | 565 | 22.1% | 1005 | 39.2% | 1570 | 56.8% |
| | Working | 229 | 8.9% | 763 | 29.8% | 992 | 43.2% |

*(Continued)*

**Table 2.** (Continued)

| Variables | Categories | Micronutrient intake status | | | | Total Weighted (n = 2562) | |
| | | No | | Yes | | | |
| | | Count | Percent | Count | Percent | Count | Percent |
|---|---|---|---|---|---|---|---|
| Sex of child | Male | 390 | 15.2% | 885 | 34.5% | 1275 | 50.1% |
| | Female | 404 | 15.8% | 883 | 34.5% | 1287 | 49.9% |
| Age of child in months | 6-12 | 422 | 16.5% | 706 | 27.6% | 1128 | 39.9% |
| | 13-23 | 372 | 14.5% | 1062 | 41.5% | 1434 | 60.1% |
| ANC follow-up | No | 364 | 14.2% | 441 | 17.2% | 805 | 24.9% |
| | Yes | 430 | 16.8% | 1327 | 51.8% | 1757 | 75.1% |
| Place of delivery | Home | 560 | 21.9% | 911 | 35.6% | 1471 | 51.5% |
| | Health facility | 234 | 9.1% | 857 | 33.5% | 1091 | 48.5% |
| PNC check-up | No | 754 | 29.4% | 1580 | 61.7% | 2334 | 89.4% |
| | Yes | 40 | 1.6% | 188 | 7.3% | 228 | 10.6% |
| Had diarrhea recently | No | 667 | 26.0% | 1442 | 56.3% | 2109 | 81.6% |
| | Yes | 127 | 5.0% | 326 | 12.7% | 453 | 18.4% |
| Had cough in last two weeks | No | 631 | 24.6% | 1416 | 55.3% | 2047 | 80.1% |
| | Yes | 163 | 6.4% | 352 | 13.7% | 515 | 19.9% |
| Media exposure | No | 588 | 23.0% | 1069 | 41.7% | 1657 | 60.5% |
| | Yes | 206 | 8.0% | 699 | 27.3% | 905 | 39.5% |

Note: others*-catholic, traditional, ANC-antenatal care, PNC-postnatal care

The line graph of incremental autocorrelation shows the minimum and the maximum distance band. The minimum distance at the beginning was 96847.00 meters (Z-score = 7.52, P-value<0.001), whereas the first maximum peak was 184351.60 meters (Z-score = 11.05, P-value<0.001) (Fig 3).

## Hot spot analysis

Based on the hotspot analysis significant clustering of micronutrient intake is detected in Addis Ababa, Harari, Dire Dawa, some parts of Gambella, and Benishangul Gumuz regions (Fig 4).

## Spatial interpolation

Ordinary kriging interpolation was computed to predict the micronutrient intake distribution among children in Ethiopia. Thus, the highest predictive prevalence of micronutrient intake increased in the Dire Dawa, Harari, Addis Ababa, Somali, Gambella, some part of Afar regions. Meanwhile, low prediction was found in the remaining areas of Ethiopia (Fig 5).

## Sat scan stastical analysis

Purely spatial analysis using the Bernoulli model was done to identify clusters with high or low micronutrient intake rates. A total of 388 significant clusters (with in six spatial windows) were identified out of the total 612 imported clusters. Among the significant clusters, there were 65 primary clusters and 323 secondary clusters. The primary clusters were located at 9.759682 N, 35.443765 E in a 203.61 km radius in Addis Ababa, Oromia, SNNPR, Gambella and some part of Benishangul Gumuz region (Fig 6).

Children who were aged 6–23 months living in the primary cluster were 29% more likely to consume foods rich in micronutrients than those outside the cluster (RR = 1.29, LLR = 25.34, P-value<0.001) (Table 3).

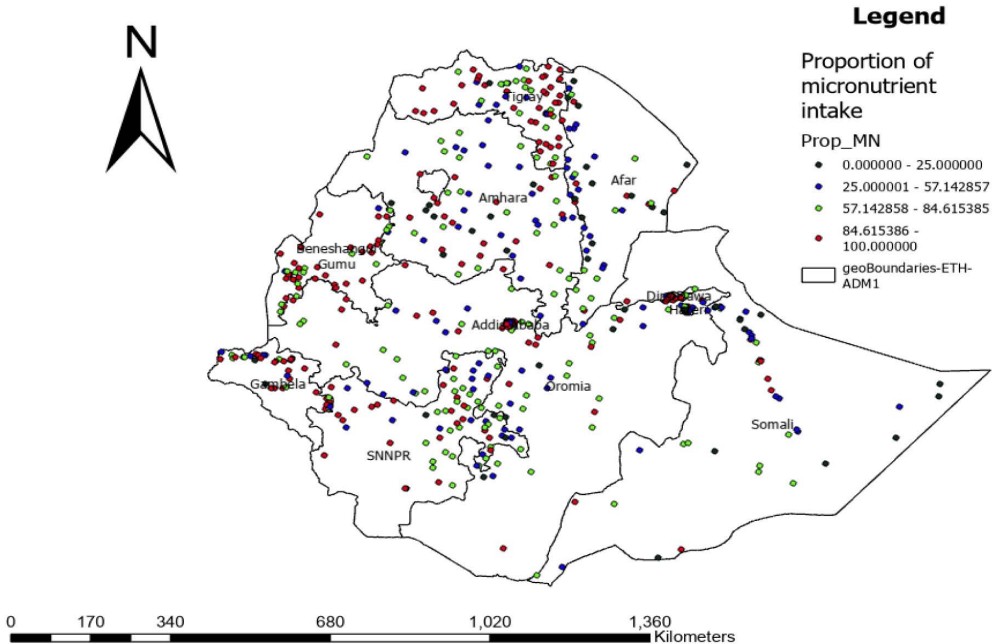

Source: Shape file from Central Stastics Agency in Ethiopia, 2013

**Fig 1. Spatial distribution of micronutrient intake among children aged 6-23 months in Ethiopia 2016.**

## Ordinary least squares (OLS) regression analysis

An ordinary least squares model was computed to identify spatial predictors of micronutrient intake. Having ANC follow-up, family size below five, rich household wealth index, child aged 13–23 months, and children who had mothers with any job were found to have positive association with micronutrient intake among children aged 6–23 months in Ethiopia. Multi-collinearity was checked by computing the Variance inflation factor (VIF). The maximum and the minimum VIF were 3.274 and 1.033, respectively. Thus, there is no significant multicollinearity between explanatory variables. Joint F-Statistic and Joint Wald Statistic result (P-value <0.001) shows that the model is statistically significant.

On the other hand, the Jarque-Bera Statistic result (P-value <0.00) explained that the OLS is biased. Furthermore, Koenker's (BP) Statistic was found to be statistically significant (P-value<0.05). Therefore, there is a possibility of hetero-scedasticity and/or nonstationarity. Thus, the model is a good candidate for further regression analysis such as GWR and MGWR analysis (Table 4).

## The geographically weighted regression (GWR) analysis result

The GWR analysis demonstrated a significant improvement compared to the global OLS model. Specifically, the AICc value decreased from 124.99 for the OLS model to 107.62 for the GWR model, indicating a difference of 17. This suggests that the GWR model better explains the spatial variations in micronutrient intake among children aged 12–23 months in Ethiopia. Additionally, the adjusted $R^2$ for the GWR model was 0.837, which shows that the model's ability to explain micronutrient intake improved significantly. In fact, GWR enhanced the explanatory power of the OLS model by approximately 22%.

Given the z-score of 6.16813, there is a less than 1% likelihood that this clustered pattern could be the result of random chance.

**Fig 2. Spatial autocorrelation analysis of spatial distribution of micronutrient intake status among children in Ethiopia, 2016 EDHS, (weighted n = 2562).**

## The multi-scale geographically weighted regression (MGWR) analysis result

The MGWR analysis showed that there was a significant improvement over global OLS and GWR. The AICc value decreased from 107 for the GWR model to 102 in MGWR model. Besides, the adjusted $R^2$ was increased by 6% from the GWR model. Finally, the model comparison was done by comparing AIC and R-squared values for each model. As a result MGWR was better than OLS and GWR model since it has a highest adjusted R2 and smallest AICc value (Table 5).

The MGWR graph revealed that a 1% increase in the child age was associated with a 19.5% higher likelihood of taking foods rich in micronutrient, particularly in the central and southern part of Ethiopia (Addis Ababa, Gambella, and SNNPR regions). In eastern and northeastern Ethiopia (Dire Dawa, Afar, and Tigray regions), ANC follow-up was the main factor influencing the likelihood of taking foods rich in micronutrient. As the mothers had ANC follow-up, the probability of micro-nutrient intake rose by 26.06%. Furthermore, the household wealth index was the primary factor influencing the likelihood of micronutrient intake. For children from wealthier households, the probability of consuming micronutrient-rich foods

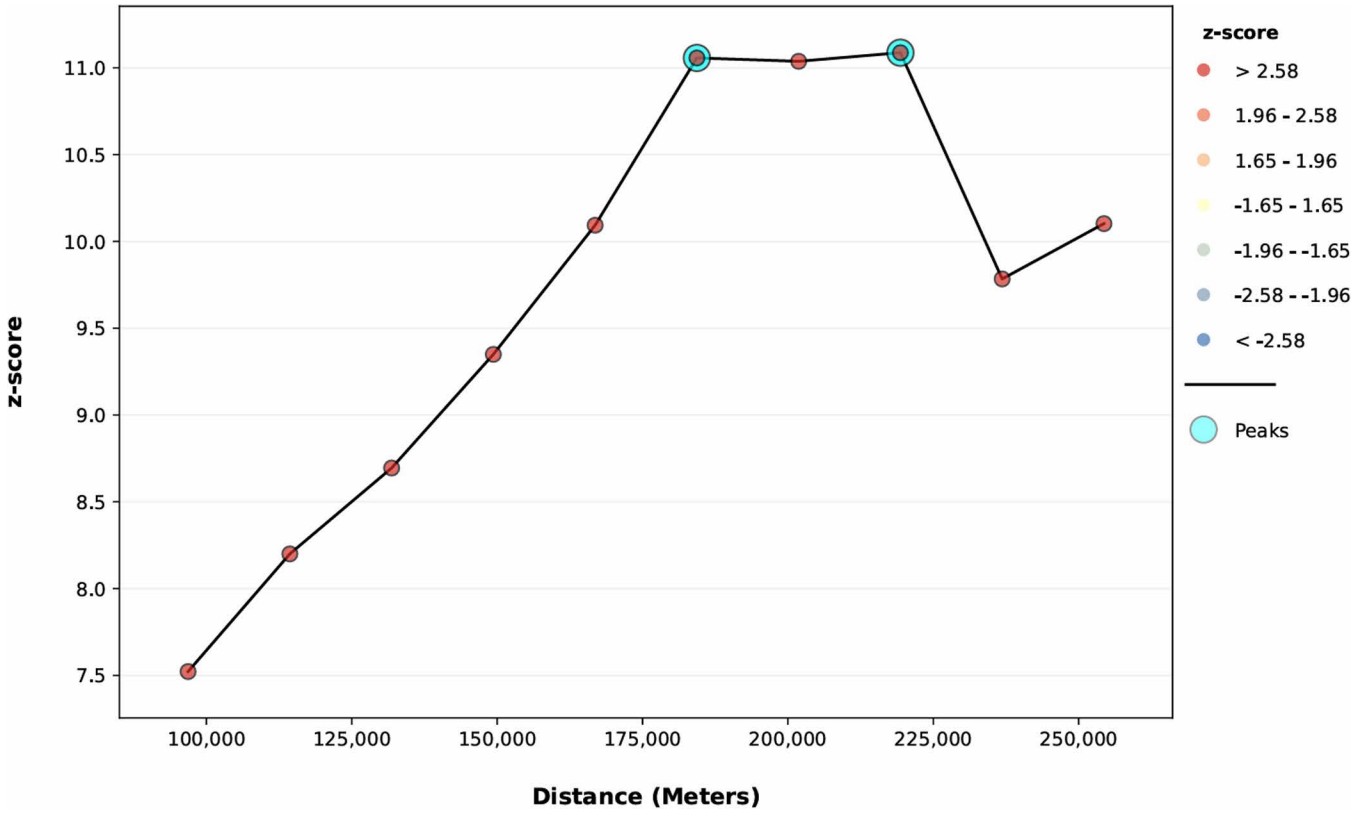

**Fig 3. Incremental autocorrelation analysis of spatial distribution of micronutrient intake status among children in Ethiopia, 2016 EDHS, (weighted n = 2562).**

increased by 3.1%. Additionally, when controlling for other factors, children whose mothers were employed were 11.25% more likely to have adequate micronutrient intake, particularly in the northeastern and southern regions of Ethiopia (Fig 7).

## Discussion

Micronutrient deficiency is a significant public health issue, especially in low and middle-income countries including Ethiopia. This study showed that only 69% (95% CI: 60.87%, 77.43%) of children aged 6–23 months had an intake of foods rich in micronutrients in Ethiopia. This result is in line with a study done in Sub-Saharan Africa (73.99%) [30] and higher than a studies done in Bangladesh [5], and South Africa [31]. These discrepancies may be due to regional differences in culture, beliefs, traditions, availability and accessibility, and religious differences. The spatial distribution of micronutrient intake of children aged 6–23 months was non-random in Ethiopia. A high (Hot spots) intake status of foods rich in micronutrient were observed in Addis Ababa, Harari, Dire Dawa, Gambella, and Benishangul Gumuz regions. In line with these regions, a spatial scan statistics analysis showed that 65 primaries (most likely) clusters with the highest micronutrient intake were identified. This result is supported by the Ethiopia National Food Consumption Survey report [32]. This might be due to urban areas typically had higher income levels and better market access, leading to greater availability of nutrient-rich foods. Families in these areas often have more purchasing power, access to a diverse range of nutrients, and higher education levels, enhancing awareness of the importance of diversified nutrition for child development. In contrast, the agrarian and hunter communities of Benishangul Gumuz and Gambella have access to a variety of nutrient sources, including both vegetables and animal products rich in micronutrients for their children.

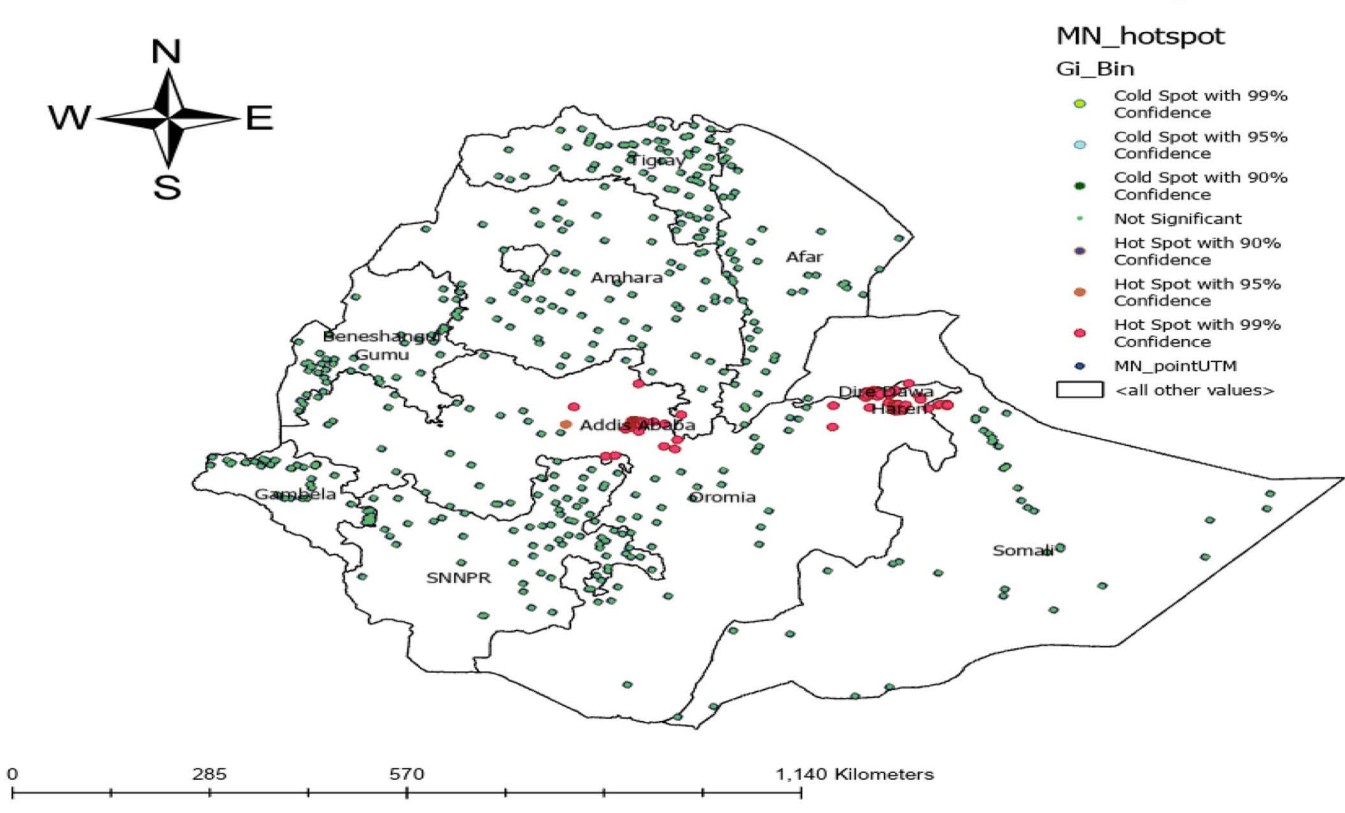

Source: Shape file from Central Stastics Agency in Ethiopia, 2013

**Fig 4. The Getis Ord Gi statistical analysis of hot spots of micronutrient intake among children in Ethiopia, EDHS 2016, (weighted n = 2562).**

A multi-scale geographically weighted regression analysis result showed that a family with a rich wealth index, child aged 13–23 months, mothers who have ANC follow-up, and children of mothers who have a job or work were a positively associated significant predictor of MN intake.

Children from wealthier families have a higher probability of taking micronutrients than their counterparts. Research has consistently shown that children from wealthier households tend to have better access to a diverse and nutrient-rich diet compared to children from poorer households [3,30,33–35]. This is because wealthier households have greater purchasing power and resources to afford a variety of foods, including fruits, vegetables, animal-source foods, and fortified foods, which are important sources of essential vitamins and minerals. Children from poorer households face food insecurity and limited access to nutritious foods, leading to micronutrient deficiencies [36]. Inadequate sanitation, poor hygiene, and limited healthcare access further exacerbate these issues. Addressing socioeconomic inequalities through targeted interventions such as food supplementation programs, nutrition education, and improving food security is crucial for reducing deficiencies. Broader policies to alleviate poverty and enhance healthcare access can help narrow the gap in micronutrient intake among children from different socioeconomic backgrounds [35].

The finding revealed that mothers who received ANC follow-up were associated with a 26.06% increase in the probability of their children achieving adequate micronutrient intake. The result is supported by other studies [30,37,38]. This

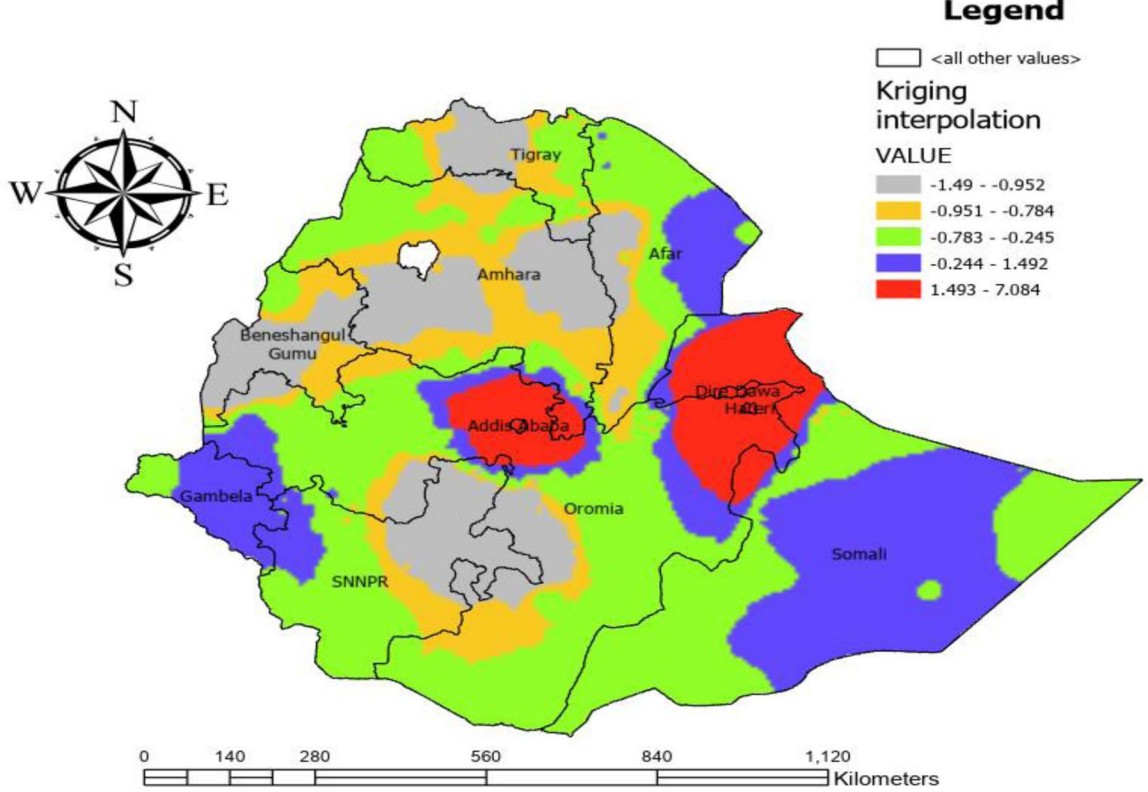

Source: Shape file from Central Statics Agency in Ethiopia, 2013

**Fig 5. Ordinary kriging interpolation to predict the prevalence of micronutrient intake among children in Ethiopia, 2016 EDHS (weighted n = 2562).**

is because ANC provides an opportunity for healthcare providers to assess and address health education and behavior change for mothers to be informed about optimal infant feeding practices, dietary diversity, and the importance of micronutrient-rich foods for child growth and development. The result implies that empowering mothers, who are on ANC follow-up with knowledge and skills related to nutrition, can help improve feeding practices and micronutrient intake among children. Strategies may include strengthening health systems to ensure equitable access to ANC, implementing community-based outreach programs to reach marginalized populations, integrating nutrition education into ANC curricula, and promoting collaboration between healthcare providers and community health workers, to deliver comprehensive maternal and child health services [38].

The result indicates a 19.15% increase in the probability of micronutrient intake as a child's age ranges from 13 to 23 compared to 6–12 month's age. This result is in congruent with many other studies conducted so far [3,30,37,39]. This might be due to developmental milestones in children, such as increased appetite and improved motor skills for self-feeding, enhance their acceptance of diverse solid foods, helping them meet micronutrient requirements. As children grow from 13 to 23 months, caregivers often become more skilled in providing balanced meals, improving micronutrient intake. Conversely, children aged 6–12 months may struggle to start complementary foods due to behavioral and cultural practices. Public health initiatives should focus on educating caregivers, promoting breastfeeding and appropriate

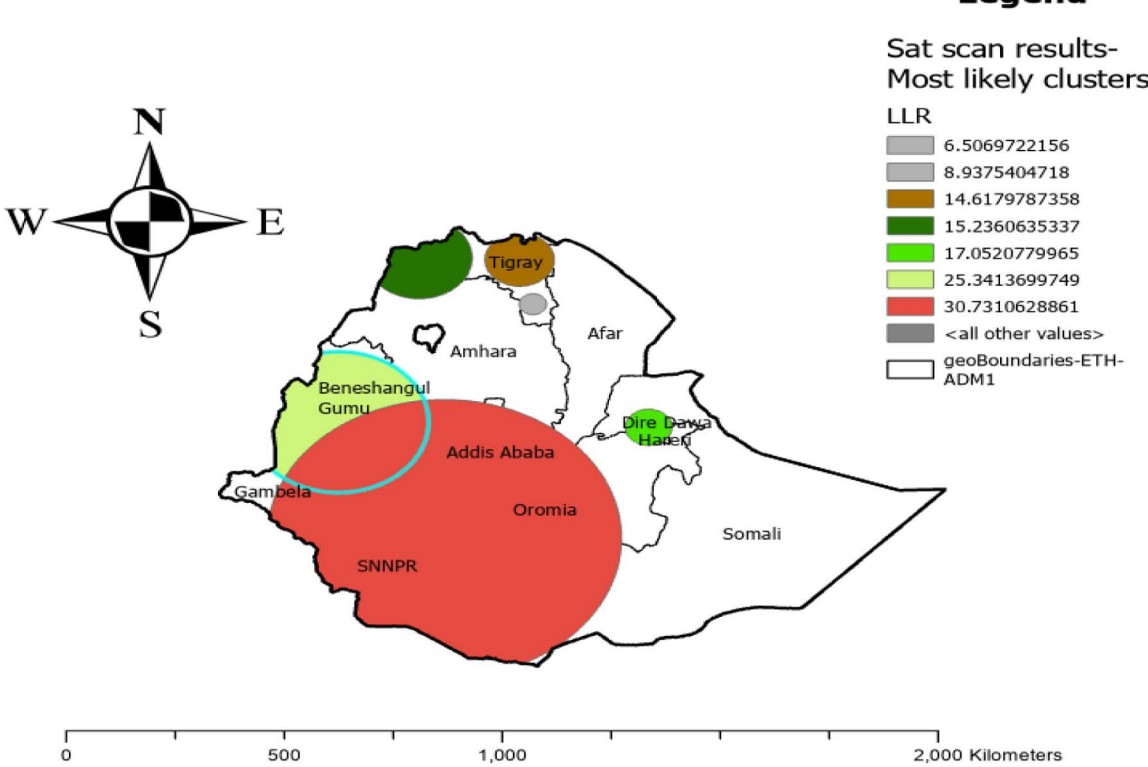

**Legend**

Sat scan results-
Most likely clusters

LLR

| | |
|---|---|
| ▨ | 6.5069722156 |
| ▨ | 8.9375404718 |
| ▨ | 14.6179787358 |
| ▨ | 15.2360635337 |
| ▨ | 17.0520779965 |
| ▨ | 25.3413699749 |
| ▨ | 30.7310628861 |
| ▨ | <all other values> |
| ☐ | geoBoundaries-ETH-ADM1 |

Source: Shape file from Central Stastics Agency in Ethiopia, 2013

**Fig 6. SaTScan scan statistics of micronutrient intake among children in Ethiopia, 2016 EDHS (weighted n = 2562).**

complementary feeding, and addressing barriers to nutrient-rich foods. Targeting interventions during this critical growth period can ensure adequate nutrition for optimal health and well-being [39].

Children who had mothers with a job or work were 11.25% more likely to have an adequate intake of micronutrients. The result is supported by other studies [3,30,33]. The possible justification may be that mothers who are employed have greater financial resources to purchase a diverse range of nutrient-rich foods for their families. Additionally, maternal employment may contribute to a more structured mealtime routine and increased access to childcare services, which can facilitate consistent and nutritious feeding practices.

Research has shown that serum measurements provide a direct assessment of nutrient levels in the body, reflecting the actual physiological status of micronutrients. In contrast, dietary recall methods, including questionnaires, rely on self-reported data that can be influenced by various factors such as recall bias and individual perceptions of food intake. Therefore, while our questionnaire serves as a practical tool for collecting data, it is important to recognize its limitations in comparison to serum measurements.

To strengthen our findings, we recommend that future studies consider a mixed-method approach that incorporates both questionnaire data and serum measurements. This would allow for a more comprehensive understanding of NM intake and its implications for health outcomes. Additionally, discussing the correlation between questionnaire results and serum levels in our findings would provide further validation for the method used.

**Table 3. Significant spatial scan statistics clusters of micronutrient intake among children aged 6–23 months, EDHS, 2016.**

| Cluster type | Detected cluster | Coordinate/ Radius | Pop-ula-tion | Cases | RR | LLR | P-value |
|---|---|---|---|---|---|---|---|
| Most likely cluster | 621, 124, 88, 349, 65, 335, 304, 320, 569, 17, 563, 416, 209, 433, 374, 244, 407, 6, 595, 462, 184, 137, 183, 203, 175, 317, 409, 165, 395, 581, 150, 248, 508, 161, 193, 275, 457, 36, 643, 558, 559, 294, 285, 246, 533, 494, 498, 615, 411, 256, 555, 280, 515, 63, 47, 469, 549, 291, 114, 386, 221, 231, 448, 541, 106 | (9.759682 N, 35.443765 E)/ 203.61 km | 269 | 234 | 1.29 | 25.34 | 0.000000027** |
| 1st Secondary | 604, 481, 355, 461, 226, 45, 129, 84, 598, 341, 404, 430, 81, 89, 636, 590, 579, 156, 623, 575, 196, 220, 479, 400, 99, 94, 298, 237, 550, 117, 597, 192, 413, 605, 538, 384, 103, 424 | (13.989017 N, 39.178052E)/ 77.13 km | 158 | 135 | 1.30 | 14.62 | 0.00037 |
| 2nd secondary cluster | 140, 546, 363, 467, 644, 43, 606, 224, 385, 390, 111, 282, 27, 493, 185, 444, 514, 5, 631, 471, 74, 380, 613, 352, 202, 25, 30, 311, 594, 166, 557, 441, 281, 523, 242, 642, 473, 453, 610, 383, 329, 173, 443, 495, 238, 381, 393, 288, 60, 614, 396, 607, 28, 228, 56, 397, 157, 419, 357, 179, 257, 387, 115, 29, 534, 418, 500, 240, 133, 321, 587, 194, 483, 580, 307, 68 | (9.624260 N, 41.839276 E)/ 52.60 km | 275 | 222 | 1.27 | 17.05 | 0.000026 |
| 3rd secondary cluster | 180, 20, 141, 53, 434, 162, 126, 565, 347, 113, 505, 406, 503, 306, 388, 373, 360, 408, 215, 148, 41, 450, 216, 634, 609, 32, 420, 331, 445, 537, 600, 86, 232, 308, 574, 182, 272, 227, 297, 12, 502, 313, 447, 576, 486, 578, 468, 365, 577, 338, 342, 316, 470, 50, 21, 405, 271, 76, 398, 589, 142, 14, 518, 174, 359, 633, 34, 432, 26, 87, 154, 489, 522, 466, 204, 207, 262, 619, 54, 422, 62, 139, 477, 438, 586, 217, 376, 325, 562, 118, 23, 177, 437, 524, 213, 485, 168, 371, 552, 459, 234, 243, 299, 290, 554, 147, 526, 197, 119, 353, 83, 46, 149, 337, 261, 475, 145, 608, 123, 645, 539, 451, 225, 110, 487, 61, 59, 195, 399, 293, 31, 107, 635, 339, 19, 108, 11, 626, 330, 305, 414, 369, 211, 582, 155, 639, 15, 274, 532, 170, 428, 247, 153, 463, 509, 125, 560, 144, 112, 464, 402, 517, 287, 90, 319, 326, 303, 601, 289, 40, 472, 377, 452, 121, 529, 423, 219, 446, 245, 593, 270, 265, 122, 218, 71, 284, 394, 572, 350, 82, 417, 201, 531, 105, 510, 7, 51, 343, 567 | (6.720108 N, 37.624880 E)/ 403.23 km | 818 | 598 | 1.32 | 30.73 | 0.00000000034 |
| 4th secondary cluster | 253, 504, 612, 296, 258, 583, 268, 78, 98, 255, 528 | (14.033877 N, 37.105922 E)/ 118.26 km | 54 | 48 | 1.67 | 15.24 | 0.000065 |
| 5th secondary cluster | 143, 449, 392, 136, 128, 79 | (12.830425 N, 39.451846 E)/ 30.51 km | 14 | 14 | 1.91 | 8.94 | 0.03 |

Note: RR-relative risk, LLR-log likelihood ratio, N-north, E-east

## Strength and limitations of the study

The utilization of nationally representative data in this study contributes to its broader applicability and statistical robustness, enabling the findings to be generalized at a national level. Furthermore, the incorporation of SaTScan and spatial distribution analysis offers valuable information regarding the geographic patterns of micronutrient intake.

However, it is important to acknowledge that the study's reliance on secondary data imposes limitations on the ability to explore the underlying causes of this phenomenon comprehensively. The data may not encompass all the factors that contribute to micronutrient intake such as seasonal variations in food availability or regional economic disparities, thereby potentially overlooking important variables. In addition, geo-scrambling of coordinates if not accounted for in the current

**Table 4. Summary of Ordinary Least Squares regression result.**

| Variables | Coefficients | Robust standard error | Robustt-t statistics | Robust probability | VIF |
|---|---|---|---|---|---|
| Intercept | 0.442 | 0.059 | 7.522 | 0.000*** | |
| Rural residence | 0.068 | 0.039 | 1.737 | 0.083 | 2.984 |
| Higher educational status of mother | 0.066 | 0.071 | 0.936 | 0.349 | 1.343 |
| Family size below five | 0.054 | 0.044 | 1.228 | 0.0219** | 1.138 |
| Rich Household Wealth index | 0.024 | 0.047 | 0.502 | 0.045** | 3.274 |
| Had media exposure | 0.032 | 0.052 | 0.619 | 0.536 | 2.407 |
| Had ANC follow-up | 0.264 | 0.043 | 6.061 | 0.000*** | 1.406 |
| Child age 13–23 months | 0.183 | 0.047 | 3.891 | 0.001*** | 1.033 |
| Had working/job | 0.127 | 0.037 | 3.478 | 0.005** | 1.105 |
| **Ordinary least square regression diagnostics** | | | | | |
| Number of Observations | 588 | AIC | | 124.99 | |
| Multiple R-Squared | 0.654 | Adjusted R-Squared | | 0.612 | |
| Joint F-Statistic | 17.237 | Prob(>F), (8,579) degree of freedom | | 0.000** | |
| Joint Wald Statistic | 127.242 | Prob(>chi-squared), (8) degree of freedom | | 0.000** | |
| Koenker's (BP) Statistic | 31.96 | Prob(>chi-squared), (8) degree of freedom | | 0.000095** | |
| Jarque-Bera Statistic | 46.75 | Prob(>chi-squared), (2) degree of freedom | | 0.000** | |

Note:

**significance

**Table 5. Summary of multi-scale geographically weighted regression analysis result and model comparisons.**

| Variables | Mean | Standard deviation | Minimum | Maximum | P-value |
|---|---|---|---|---|---|
| Intercept | 0.0074 | 0.2088 | -0.5911 | 0.3852 | |
| Rural residence | 0.1003 | 0.0039 | 0.0937 | 0.1129 | 0.126 |
| Higher educational status | 0.0253 | 0.0029 | 0.0183 | 0.0342 | 0.266 |
| Family size less than five | 0.0478 | 0.1024 | 0.0174 | 0.3402 | 0.141 |
| Rich Wealth index | 0.0312 | 0.0057 | 0.0031 | 0.0484 | 0.026** |
| Had media exposure | 0.0394 | 0.0977 | 0.0220 | 0.2785 | 0.060 |
| Had ANC follow-up | 0.2606 | 0.0662 | 0.1587 | 0.3878 | 0.000*** |
| Child age 13–24 months | 0.1915 | 0.0622 | 0.1207 | 0.3108 | 0.000*** |
| Had working/job | 0.1125 | 0.0164 | 0.0723 | 0.1393 | 0.000*** |
| **Model comparison (OLS vs. GWR vs. MGWR)** | | | | | |
| Parameters | OLS model | | GWR model | MGWR model | |
| AICc | 124.98 | | 107.616 | 102.34 | |
| R-Squared | 0.654 | | 0.878 | 0.916 | |
| Adjusted R-Squared | 0.612 | | 0.837 | 0.891 | |

study and the measurement of the consumption status of micronutrient-rich foods through a 24-hour recall approach introduces the possibility of recall, social desirability biases, and does not represent the usual dietary behavior. Furthermore, we acknowledge that supplementation often occurs in response to a diagnosed deficiency, which can complicate our analysis of the relationship between MN intake and health outcomes.

To address this concern, we would like to highlight the importance of longitudinal studies to better establish cause-and-effect relationships in future research.

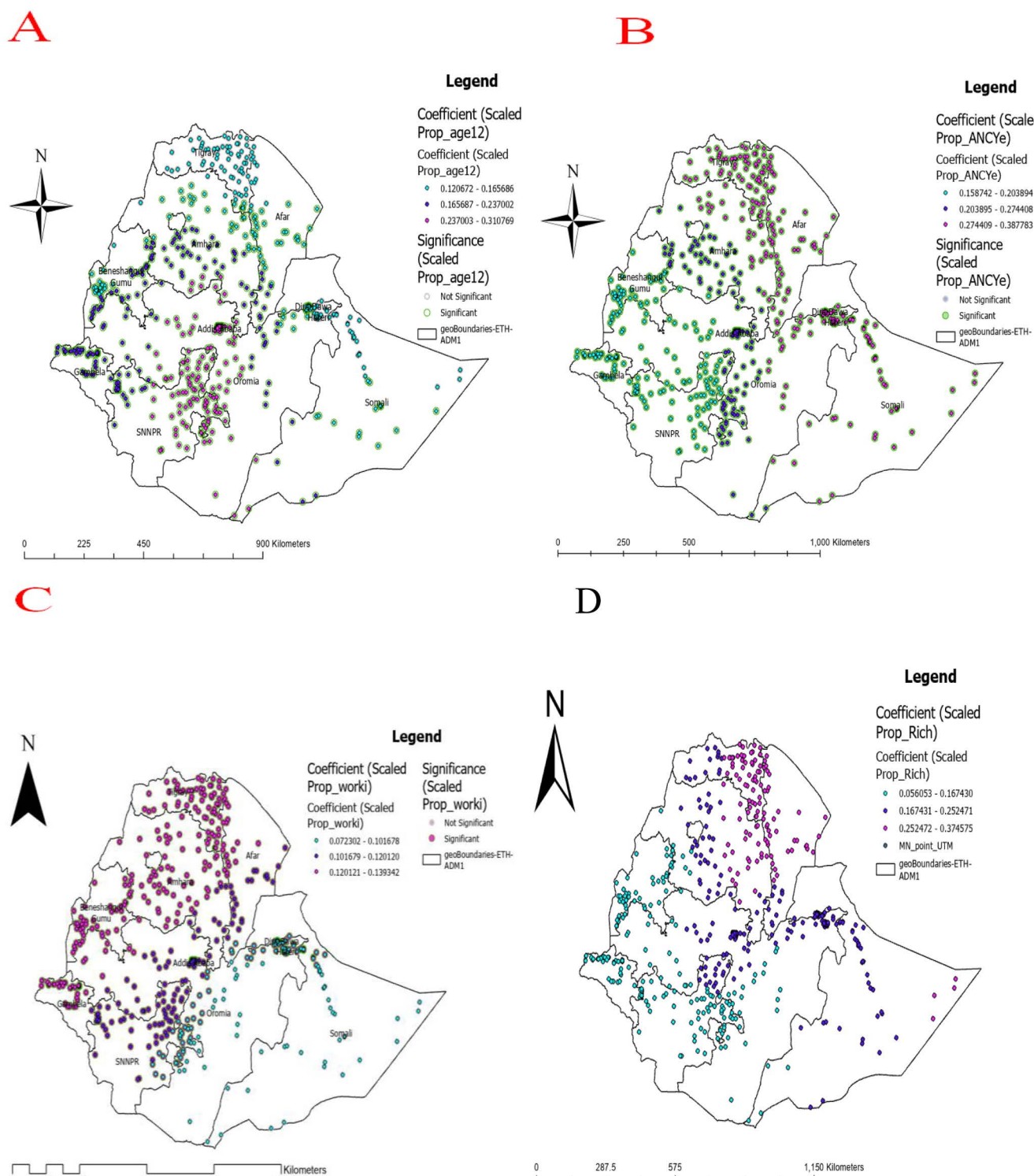

**Fig 7. The spatial mapping of multi-scale geographically weighted regression coefficients by child aged 13-23 months (A), ANC follow-up (B), mother having job/work (C), and household with rich wealth index (D) to predict the hotspot of micronutrient intake among children in Ethiopia.**

## Conclusion and implication of the study

In Ethiopia, the intake of foods rich in MN varies across different regions. Through spatial analysis, statistically significant hot spots of high intake of these nutrient-rich foods were identified in Addis Ababa, Harari, Dire Dawa, certain parts of Benishangul, and the Gambella region.

This study revealed that the factors influencing the intake of MN-rich foods differ spatially throughout Ethiopia. Specifically, household wealth status, child aged 13–23 months, the mother's occupation, and the utilization of ANC services were found to be statistically significant predictors in various regions of Ethiopia.

These findings highlight the importance of targeted nutrition intervention programs in the identified areas to improve the intake of nutrient-rich foods among children. Encouraging ANC service utilization, providing occupation, and enhancing household wealth index are particularly recommended strategies. By addressing these contributing factors in the identified geographic locations, significant improvements in child nutrition and optimal growth can be achieved.

It should be noted that this study focused specifically on the consumption of foods rich in MN, and the association between the dependent and independent variables may be limited due to the use of secondary data. Future research should consider utilizing different datasets to assess other relevant factors associated with micronutrient intake among children, especially in areas with high levels of micronutrient insecurity.

## Acknowledgments

The authors are indebted to the DHS program for giving permission to access the dataset.

## Author contributions

**Conceptualization:** Alemu Birara Zemariam.

**Data curation:** Alemu Birara Zemariam, Addis Wondmagegn Alamaw, Ribka Nigatu Haile.

**Formal analysis:** Alemu Birara Zemariam.

**Methodology:** Alemu Birara Zemariam, Tesfaye Engdaw Habtie.

**Software:** Alemu Birara Zemariam.

**Validation:** Alemu Birara Zemariam.

**Writing – original draft:** Alemu Birara Zemariam, Addis Wondmagegn Alamaw, Rediet Woldesenbet Molla, Tesfaye Engdaw Habtie, Molla Azmeraw Bizuayehu, Ribka Nigatu Haile, Tegene Atamenta Kitaw, Biruk Beletew Abate, Mollalign Aligaz Adisu.

**Writing – review & editing:** Alemu Birara Zemariam, Mollalign Aligaz Adisu.

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
