## [Decision Letter · Decision Letter 0]

4 Oct 2024

PONE-D-24-22661Spatial distribution and determinants of micronutrient intake status among children aged 6-23 months in Ethiopia: A Multi-scale Geographical Weighted Regression AnalysisPLOS ONE

Dear Dr. Zemariam,

Thank you for submitting your manuscript to PLOS ONE. After careful consideration, we feel that it has merit but does not fully meet PLOS ONE’s publication criteria as it currently stands. Therefore, we invite you to submit a revised version of the manuscript that addresses the points raised during the review process.

The reviewers' comments are as follows;

**Reviewer 1**

Need to provide more information about the location (coordinates) geo-scrambling/dithering; why it’s necessary, how it affects your analysis and how you mitigated for it in the analysisYou need a summary section of all the methods which may be interconnected with a clear simple figure, to show what feeds where and why all the methods are necessary. It seems hotspot analysis, SaTScan analysis and spatial prediction(kriging) all point towards the same message. Therefore this summary paragraph early on will be useful for the readers.Also, note that the values of zero indicate perfect randomness.Likewise, provide a preamble to local Getis Ord statistics for the readerOn Spatial interpolation, need to use a variogram to test the spatial dependence for micronutrients intake including details of practical range.What covariates were used for spatial prediction of short birth interval? These need to be specified and included. Also, put down the equation for easier comprehension. State-of-the-art model-based geostatistics would be a better fit to fully exploit the posterior samples than kriging. Therefore justify your choice of the modelling approachMultilevel modelling should maybe come in before the spatial prediction so that the covariates are included during spatial prediction for robust resultsOn Statistical analysis, the statement “Finally, the mixed effect model, which included both fixed and random effect variables were fitted. To include the variable in the model p-value.”; seems to be incomplete and in the wrong placeOn SaTScan analysis, please state the unit of analysis. Does the SaTScan analysis and hotspot analysis provide similar insights? Why was it important to have these two in the paper?On Multilevel binary logistic analysis, why was clustering considered only at the cluster level? Need to account for the fact that individuals are within households, households within clusters and clusters within a regionThe discussion has a big section which reads like a repetition of the results section. Please edit this section to include details on what this study means to Ethiopia and the global community in terms of policy and actions that can be takenGiven the importance of subnational targeting, why were districts and/or region-level estimates not presented? These would align with units of decision-making and can be obtained by aggregating results of spatial prediction. The single national value masks heterogeneity while the pixel map is not used as a policy actionable unitUnder limitations- include geo-scrambling of coordinates if not accounted for in the analysisThe document needs language editionYou have published with other method of analysis? So would you think that it will provides another insights? I need a clear answer. Unless biased.

**Reviewer 2**

**Introduction**

The introduction effectively highlights the global and national impact of micronutrient (MN) deficiency, particularly among children in Ethiopia. However, it could be improved by refining its structure, enhancing clarity, and avoiding repetition. Here are some suggestions:

1. It would be better to add a specialized definition of malnutrition at the beginning of the introduction.

2. For improving the coherence and clarity of the introduction section, you can proceed based on the following breakdown and provide an explanation.

a. In the first paragraph, you can briefly explain the following topics mentioned in the introduction: MN deficiency as a global public health concern, the importance of the first 1,000 days of a child's life, global scale of malnutrition, and vulnerability in Low- and middle-income countries.

b. In the second paragraph, you can discuss the importance of adequate micronutrient intake, provide definitions and classifications of micronutrients, and address other related issues concerning the mentioned micronutrients.

c. In the third paragraph, discuss strategies related to reducing malnutrition, the recommendations from the World Health Organization, the effectiveness of supplementation programs, and factors influencing malnutrition.

d. Finally, identify the existing gaps, and state your objective

**Method**

The methods section is thorough and detailed but could benefit from improved organization. Here are some suggestions:

1. Divide the methods section into subsections such as "Data Source," "Study Population," "Variables and Measurements," "Data Processing and Analysis," and "Model Evaluation" for better readability.

2. Use a reporting guideline suited to your article to better organize headings and enhance clarity and order.

3. Depending on the authors' preferences, explain why specific techniques, like Multi-Scale Geographically Weighted Regression (MGWR), were chosen over others, such as Geographically Weighted Regression (GWR). Providing a rationale for the superiority of MGWR could help clarify its advantages.

**Results**

The results section offers a detailed and data-rich summary but could be improved for clarity and precision:

1. Amend the grammatical error in the first line from "More than one-third (34.5%) of the female children were had taken foods rich in micronutrient" to "More than one-third (34.5%) of the female children had consumed foods rich in micronutrients."

2. It might be beneficial to summarize key findings more clearly. For instance, explicitly state the regions with the highest and lowest micronutrient intake before discussing the methods used.

**Discussion**

The discussion is generally well-structured and comprehensive but could be enhanced by improving clarity, conciseness, and emphasis:

1. Shorten long sentences for better clarity. For example, sentence 33 starting with "Addressing socioeconomic inequalities..." could be split into two sentences.

2. Avoid repeating explanations about previously discussed points, such as the role of wealth in influencing micronutrient intake. Consolidate these discussions for brevity.

**Limitations**

1. While limitations are acknowledged, they could be expanded to address potential confounders or biases not controlled for in the study. Discuss factors like seasonal variations in food availability or regional economic disparities as additional limitations.

**Reviewer 3**

The manuscript details an interesting piece of research, and seems technically sound. "Hidden hunger" is a key public and global health concern, especially in infants in lower-income countries, and in light of climate change pressurizing the situation. The statistical analysis is complex and thorough, and appropriate to answer the research questions. The conclusions are the same key messages that the reader draws from the data. There are a few clarifications that I would like the authors to make, as well as some grammatical areas I have highlighted and commented on the attachment. All of my comments can be read on the PDF. Overall, a very good study and clearly written paper.

**Reviewer 5**

This relevant work aimed to assess the spatial distribution and factors influencing the intake of foods rich in MN among children aged 6–23 months in Ethiopia. The study has national representativeness and a robust statistical analysis.

Below are some minor and major comments:

**Minor comments:**

The text does not define the meaning of ANC, nor how the wealth index variable is constructed

Regarding table 1:

Why not evaluate age as a continuous variable?

Sex of the household? (word “head” is missing)

Family size is repeated

To include AIC in the second part of Table 4? If I understand correctly, the AIC is a criterion to select between more than one model, what would be the use of adding it in isolation?

**Major comments:**

Some variables of nutritional status in the infant. Is it correct to assume that infants who have not consumed MN in the last 24 hours have MN deficiency? I think that is not an adequate assumption, it could be possible that they do not have MN deficiency even if they have not consumed MN in the previous 24 hours. Did you take any consideration regarding breastfeeding? Breastfeeding could also provide NM to the infant. If you do not have access to this information, you could at least develop this point in the discussion.

Nutrient intake was assessed by 24-hour recall, and the day of the week was indicated. If this survey was conducted on a Monday, it is possible that the previous day is not representative of usual dietary behaviour. This is an important limitation that should be addressed in the discussion.

I think the discussion could include something about the validation of the method to assess NM intake, perhaps how close the questionnaire is to the gold standard (serum measurement) this argument would give strength to the discussion.

The discussion is developed around the purchasing power of families and therefore the ability to acquire foods rich in NM. However, there is a lack of information about the food environment in the centre of urban areas, and how this could be promoting or not the consumption of foods rich in NM (commercial determinants of nutrition).

I think the study includes many results and some of them evaluate the same associations, not all the results are discussed, you should consider sending some to supplementary material.

You could mention in the introduction or discussion how spatial distribution studies have been useful in planning public health and nutrition intervention projects.

An important issue to consider in the discussion is the possible reverse causality in children who consume MN supplements since they usually do so when they already have a diagnosis of deficiency.

We look forward to receiving your revised manuscript.

Kind regards,

Samuel Kofi Tchum, Ph.D.

Academic Editor

PLOS ONE

Journal Requirements:

3. We note that [Figures 1,4,5,6 and 7] in your submission contain [map/satellite] images which may be copyrighted. All PLOS content is published under the Creative Commons Attribution License (CC BY 4.0), which means that the manuscript, images, and Supporting Information files will be freely available online, and any third party is permitted to access, download, copy, distribute, and use these materials in any way, even commercially, with proper attribution. For these reasons, we cannot publish previously copyrighted maps or satellite images created using proprietary data, such as Google software (Google Maps, Street View, and Earth). For more information, see our copyright guidelines: http://journals.plos.org/plosone/s/licenses-and-copyright.

a. You may seek permission from the original copyright holder of Figures 1,4,5,6 and 7 to publish the content specifically under the CC BY 4.0 license. 

Additional Editor Comments:

Reviewer 1

1. Need to provide more information about the location (coordinates) geo-scrambling/dithering; why it’s necessary, how it affects your analysis and how you mitigated for it in the analysis

2. You need a summary section of all the methods which may be interconnected with a clear simple figure, to show what feeds where and why all the methods are necessary. It seems hotspot analysis, SaTScan analysis and spatial prediction(kriging) all point towards the same message. Therefore this summary paragraph early on will be useful for the readers.

3. Also, note that the values of zero indicate perfect randomness.

4. Likewise, provide a preamble to local Getis Ord statistics for the reader

5. On Spatial interpolation, need to use a variogram to test the spatial dependence for micronutrients intake including details of practical range.

6. What covariates were used for spatial prediction of short birth interval? These need to be specified and included. Also, put down the equation for easier comprehension. State-of-the-art model-based geostatistics would be a better fit to fully exploit the posterior samples than kriging. Therefore justify your choice of the modelling approach

7. Multilevel modelling should maybe come in before the spatial prediction so that the covariates are included during spatial prediction for robust results

8. On Statistical analysis, the statement “Finally, the mixed effect model, which included both fixed and random effect variables were fitted. To include the variable in the model p-value.”; seems to be incomplete and in the wrong place

9. On SaTScan analysis, please state the unit of analysis. Does the SaTScan analysis and hotspot analysis provide similar insights? Why was it important to have these two in the paper?

10. On Multilevel binary logistic analysis, why was clustering considered only at the cluster level? Need to account for the fact that individuals are within households, households within clusters and clusters within a region

11. The discussion has a big section which reads like a repetition of the results section. Please edit this section to include details on what this study means to Ethiopia and the global community in terms of policy and actions that can be taken

12. Given the importance of subnational targeting, why were districts and/or region-level estimates not presented? These would align with units of decision-making and can be obtained by aggregating results of spatial prediction. The single national value masks heterogeneity while the pixel map is not used as a policy actionable unit

13. Under limitations- include geo-scrambling of coordinates if not accounted for in the analysis

14. The document needs language edition

15. You have published with other method of analysis? So would you think that it will provides another insights? I need a clear answer. Unless biased.

Reviewer 2

Introduction

The introduction effectively highlights the global and national impact of micronutrient (MN) deficiency, particularly among children in Ethiopia. However, it could be improved by refining its structure, enhancing clarity, and avoiding repetition. Here are some suggestions:

1. It would be better to add a specialized definition of malnutrition at the beginning of the introduction.

2. For improving the coherence and clarity of the introduction section, you can proceed based on the following breakdown and provide an explanation.

a. In the first paragraph, you can briefly explain the following topics mentioned in the introduction: MN deficiency as a global public health concern, the importance of the first 1,000 days of a child's life, global scale of malnutrition, and vulnerability in Low- and middle-income countries.

b. In the second paragraph, you can discuss the importance of adequate micronutrient intake, provide definitions and classifications of micronutrients, and address other related issues concerning the mentioned micronutrients.

c. In the third paragraph, discuss strategies related to reducing malnutrition, the recommendations from the World Health Organization, the effectiveness of supplementation programs, and factors influencing malnutrition.

d. Finally, identify the existing gaps, and state your objective

Method

The methods section is thorough and detailed but could benefit from improved organization. Here are some suggestions:

1. Divide the methods section into subsections such as "Data Source," "Study Population," "Variables and Measurements," "Data Processing and Analysis," and "Model Evaluation" for better readability.

2. Use a reporting guideline suited to your article to better organize headings and enhance clarity and order.

3. Depending on the authors' preferences, explain why specific techniques, like Multi-Scale Geographically Weighted Regression (MGWR), were chosen over others, such as Geographically Weighted Regression (GWR). Providing a rationale for the superiority of MGWR could help clarify its advantages.

Results

The results section offers a detailed and data-rich summary but could be improved for clarity and precision:

1. Amend the grammatical error in the first line from "More than one-third (34.5%) of the female children were had taken foods rich in micronutrient" to "More than one-third (34.5%) of the female children had consumed foods rich in micronutrients."

2. It might be beneficial to summarize key findings more clearly. For instance, explicitly state the regions with the highest and lowest micronutrient intake before discussing the methods used.

Discussion

The discussion is generally well-structured and comprehensive but could be enhanced by improving clarity, conciseness, and emphasis:

1. Shorten long sentences for better clarity. For example, sentence 33 starting with "Addressing socioeconomic inequalities..." could be split into two sentences.

2. Avoid repeating explanations about previously discussed points, such as the role of wealth in influencing micronutrient intake. Consolidate these discussions for brevity.

Limitations

1. While limitations are acknowledged, they could be expanded to address potential confounders or biases not controlled for in the study. Discuss factors like seasonal variations in food availability or regional economic disparities as additional limitations.

Reviewer 3

• The manuscript details an interesting piece of research, and seems technically sound. "Hidden hunger" is a key public and global health concern, especially in infants in lower-income countries, and in light of climate change pressurizing the situation. The statistical analysis is complex and thorough, and appropriate to answer the research questions. The conclusions are the same key messages that the reader draws from the data. There are a few clarifications that I would like the authors to make, as well as some grammatical areas I have highlighted and commented on the attachment. All of my comments can be read on the PDF. Overall, a very good study and clearly written paper.

Reviewer 4

• Not Submitted

Reviewer 5

This relevant work aimed to assess the spatial distribution and factors influencing the intake of foods rich in MN among children aged 6–23 months in Ethiopia. The study has national representativeness and a robust statistical analysis.

Below are some minor and major comments:

Minor comments:

The text does not define the meaning of ANC, nor how the wealth index variable is constructed

Regarding table 1:

Why not evaluate age as a continuous variable?

Sex of the household? (word “head” is missing)

Family size is repeated

To include AIC in the second part of Table 4? If I understand correctly, the AIC is a criterion to select between more than one model, what would be the use of adding it in isolation?

Major comments:

Some variables of nutritional status in the infant. Is it correct to assume that infants who have not consumed MN in the last 24 hours have MN deficiency? I think that is not an adequate assumption, it could be possible that they do not have MN deficiency even if they have not consumed MN in the previous 24 hours. Did you take any consideration regarding breastfeeding? Breastfeeding could also provide NM to the infant. If you do not have access to this information, you could at least develop this point in the discussion.

Nutrient intake was assessed by 24-hour recall, and the day of the week was indicated. If this survey was conducted on a Monday, it is possible that the previous day is not representative of usual dietary behaviour. This is an important limitation that should be addressed in the discussion.

I think the discussion could include something about the validation of the method to assess NM intake, perhaps how close the questionnaire is to the gold standard (serum measurement) this argument would give strength to the discussion.

The discussion is developed around the purchasing power of families and therefore the ability to acquire foods rich in NM. However, there is a lack of information about the food environment in the centre of urban areas, and how this could be promoting or not the consumption of foods rich in NM (commercial determinants of nutrition).

I think the study includes many results and some of them evaluate the same associations, not all the results are discussed, you should consider sending some to supplementary material.

You could mention in the introduction or discussion how spatial distribution studies have been useful in planning public health and nutrition intervention projects.

An important issue to consider in the discussion is the possible reverse causality in children who consume MN supplements since they usually do so when they already have a diagnosis of deficiency.

Reviewers' comments:

Reviewer's Responses to Questions

**Comments to the Author**

1. Is the manuscript technically sound, and do the data support the conclusions?

Reviewer #1: Partly

Reviewer #2: Yes

Reviewer #3: Yes

Reviewer #4: Yes

Reviewer #5: Partly

2. Has the statistical analysis been performed appropriately and rigorously? 

Reviewer #1: Yes

Reviewer #2: I Don't Know

Reviewer #3: Yes

Reviewer #4: Yes

Reviewer #5: Yes

3. Have the authors made all data underlying the findings in their manuscript fully available?

Reviewer #1: Yes

Reviewer #2: Yes

Reviewer #3: Yes

Reviewer #4: Yes

Reviewer #5: Yes

4. Is the manuscript presented in an intelligible fashion and written in standard English?

Reviewer #1: Yes

Reviewer #2: No

Reviewer #3: Yes

Reviewer #4: Yes

Reviewer #5: Yes

5. Review Comments to the Author

Reviewer #1: 1. Need to provide more information about the location (coordinates) geo-scrambling/dithering; why it’s necessary, how it affects your analysis and how you mitigated for it in the analysis

2. You need a summary section of all the methods which may be interconnected with a clear simple figure, to show what feeds where and why all the methods are necessary. It seems hotspot analysis, SaTScan analysis and spatial prediction(kriging) all point towards the same message. Therefore this summary paragraph early on will be useful for the readers.

3. Also, note that the values of zero indicate perfect randomness.

4. Likewise, provide a preamble to local Getis Ord statistics for the reader

5. On Spatial interpolation, need to use a variogram to test the spatial dependence for micronutrients intake including details of practical range.

6. What covariates were used for spatial prediction of short birth interval? These need to be specified and included. Also, put down the equation for easier comprehension. State-of-the-art model-based geostatistics would be a better fit to fully exploit the posterior samples than kriging. Therefore justify your choice of the modelling approach

7. Multilevel modelling should maybe come in before the spatial prediction so that the covariates are included during spatial prediction for robust results

8. On Statistical analysis, the statement “Finally, the mixed effect model, which included both fixed and random effect variables were fitted. To include the variable in the model p-value.”; seems to be incomplete and in the wrong place

9. On SaTScan analysis, please state the unit of analysis. Does the SaTScan analysis and hotspot analysis provide similar insights? Why was it important to have these two in the paper?

10. On Multilevel binary logistic analysis, why was clustering considered only at the cluster level? Need to account for the fact that individuals are within households, households within clusters and clusters within a region

11 The discussion has a big section which reads like a repetition of the results section. Please edit this section to include details on what this study means to Ethiopia and the global community in terms of policy and actions that can be taken

12. Given the importance of subnational targeting, why were districts and/or region-level estimates not presented? These would align with units of decision-making and can be obtained by aggregating results of spatial prediction. The single national value masks heterogeneity while the pixel map is not used as a policy actionable unit

13. Under limitations- include geo-scrambling of coordinates if not accounted for in the analysis

14. The document needs language edition

15. You have published with other method of analysis? So would you think that it will provides another insights? I need a clear answer. Unless biased.

Reviewer #2: Introduction

The introduction effectively highlights the global and national impact of micronutrient (MN) deficiency, particularly among children in Ethiopia. However, it could be improved by refining its structure, enhancing clarity, and avoiding repetition. Here are some suggestions:

1. It would be better to add a specialized definition of malnutrition at the beginning of the introduction.

2. For improving the coherence and clarity of the introduction section, you can proceed based on the following breakdown and provide an explanation.

a. In the first paragraph, you can briefly explain the following topics mentioned in the introduction: MN deficiency as a global public health concern, the importance of the first 1,000 days of a child's life, global scale of malnutrition, and vulnerability in Low- and middle-income countries.

b. In the second paragraph, you can discuss the importance of adequate micronutrient intake, provide definitions and classifications of micronutrients, and address other related issues concerning the mentioned micronutrients.

c. In the third paragraph, discuss strategies related to reducing malnutrition, the recommendations from the World Health Organization, the effectiveness of supplementation programs, and factors influencing malnutrition.

d. Finally, identify the existing gaps, and state your objective

Method

The methods section is thorough and detailed but could benefit from improved organization. Here are some suggestions:

1. Divide the methods section into subsections such as "Data Source," "Study Population," "Variables and Measurements," "Data Processing and Analysis," and "Model Evaluation" for better readability.

2. Use a reporting guideline suited to your article to better organize headings and enhance clarity and order.

3. Depending on the authors' preferences, explain why specific techniques, like Multi-Scale Geographically Weighted Regression (MGWR), were chosen over others, such as Geographically Weighted Regression (GWR). Providing a rationale for the superiority of MGWR could help clarify its advantages.

Results

The results section offers a detailed and data-rich summary but could be improved for clarity and precision:

1. Amend the grammatical error in the first line from "More than one-third (34.5%) of the female children were had taken foods rich in micronutrient" to "More than one-third (34.5%) of the female children had consumed foods rich in micronutrients."

2. It might be beneficial to summarize key findings more clearly. For instance, explicitly state the regions with the highest and lowest micronutrient intake before discussing the methods used.

Discussion

The discussion is generally well-structured and comprehensive but could be enhanced by improving clarity, conciseness, and emphasis:

1. Shorten long sentences for better clarity. For example, sentence 33 starting with "Addressing socioeconomic inequalities..." could be split into two sentences.

2. Avoid repeating explanations about previously discussed points, such as the role of wealth in influencing micronutrient intake. Consolidate these discussions for brevity.

Limitations

1. While limitations are acknowledged, they could be expanded to address potential confounders or biases not controlled for in the study. Discuss factors like seasonal variations in food availability or regional economic disparities as additional limitations.

Reviewer #3: The manuscript details an interesting piece of research, and seems technically sound. "Hidden hunger" is a key public and global health concern, especially in infants in lower-income countries, and in light of climate change pressurizing the situation. The statistical analysis is complex and thorough, and appropriate to answer the research questions. The conclusions are the same key messages that the reader draws from the data. There are a few clarifications that I would like the authors to make, as well as some grammatical areas I have highlighted and commented on the attachment. All of my comments can be read on the PDF. Overall, a very good study and clearly written paper.

Reviewer #4: Dear Author,

I wanted to bring to your attention that I previously reviewed manuscript “PONE-D-24-22661” and provided several comments and suggestions for revision. However, in the current version, I do not see any changes that indicate my feedback was addressed.

If the revisions have been made, I would appreciate receiving a version with the changes clearly highlighted or tracked.

Best regards,

Reviewer #5: This relevant work aimed to assess the spatial distribution and factors influencing the intake of foods rich in MN among children aged 6–23 months in Ethiopia. The study has national representativeness and a robust statistical analysis.

Below are some minor and major comments:

Minor comments:

The text does not define the meaning of ANC, nor how the wealth index variable is constructed

Regarding table 1:

Why not evaluate age as a continuous variable?

Sex of the household? (word “head” is missing)

Family size is repeated

To include AIC in the second part of table 4? If I understand correctly, the AIC is a criterion to select between more than one model, what would be the use of adding it in isolation?

Major comments:

Some variables of nutritional status in the infant. Is it correct to assume that infants who have not consumed MN in the last 24 hours have MN deficiency? I think that is not an adequate assumption, it could be possible that they do not have MN deficiency even if they have not consumed MN in the previous 24 hours. Did you take any consideration regarding breastfeeding? Breastfeeding could also provide NM to the infant. If you do not have access to this information, you could at least develop this point in the discussion.

Nutrient intake was assessed by 24-hour recall, the day of the week was indicated. If this survey was conducted on a Monday, it is possible that the previous day is not representative of usual dietary behavior. This is an important limitation that should be addressed in the discussion.

I think the discussion could include something about the validation of the method to assess NM intake, perhaps how close the questionnaire is to the gold standard (serum measurement) this argument would give strength to the discussion.

The discussion is developed around the purchasing power of families and therefore the ability to acquire foods rich in NM. However, there is a lack of information about the food environment in the center of urban areas, and how this could be promoting or not the consumption of foods rich in NM (commercial determinants of nutrition).

I think the study includes many results and some of them evaluate the same associations, not all the results are discussed, you should consider sending some to supplementary material.

You could mention in the introduction or discussion how spatial distribution studies have been useful in planning public health and nutrition intervention projects.

An important issue to consider in the discussion is the possible reverse causality in children who consume MN supplements since they usually do so when they already have a diagnosis of deficiency.

6. PLOS authors have the option to publish the peer review history of their article (what does this mean? ). If published, this will include your full peer review and any attached files.

**Do you want your identity to be public for this peer review?** For information about this choice, including consent withdrawal, please see our Privacy Policy .

Reviewer #1: No

Reviewer #2: No

Reviewer #3: **Yes: ** Jessica Boxall ANutr

Reviewer #4: **Yes: ** Dr. Mai Albaik

Reviewer #5: **Yes: ** Nadia Angélica Cerecer-Ortiz

---

## [Author Response · Author response to Decision Letter 1]

14 Feb 2025

Dear Esteemed editorial team,

We appreciate your important comments. We have accepted your comment and update the data availability statement. Additionally, thank you for your feedback regarding Figures 1, 4, 5, 6, and 7. We generated these maps using ArcGIS Pro software and assure you that all figures are original, based on the GPS data we accessed from the following websites. We mistakenly attached the incorrect link and apologize for the inconsistency. We used the Africa Geo Portal for the previously published article entitled “Spatial Variation, 20-Year Trends, and Determinants of the Double Burden of Wasting and Stunting Among Under-Five Children in Ethiopia: A Geo-Spatial and Multivariate Decomposition Analysis (2000–2019) and Mapping Fertility Rates at National, Sub-National, and Local Levels in Ethiopia Between 2000 and 2019,” but it was not used in this study. Therefore, all of the figures listed above contain map data retrieved from the DHS Program through the link below.

https://spatialdata.dhsprogram.com/data/#/

We appreciate your guidance in this matter and are committed to ensuring our submission meets all necessary requirements.

Thank you once again for your assistance.

---

## [Editor Report · Decision Letter 1]

3 Mar 2025

Spatial distribution and determinants of micronutrient intake status among children aged 6-23 months in Ethiopia: A Multi-scale Geographical Weighted Regression Analysis

PONE-D-24-22661R1

Dear Dr. Alemu Birara Zemariam,

We’re pleased to inform you that your manuscript has been judged scientifically suitable for publication and will be formally accepted for publication once it meets all outstanding technical requirements.

Kind regards,

Samuel Kofi Tchum, Ph.D.

Academic Editor

PLOS ONE
---

## [Editor Report · Acceptance letter]

PONE-D-24-22661R1

PLOS ONE

Dear Dr. Zemariam,

I'm pleased to inform you that your manuscript has been deemed suitable for publication in PLOS ONE. Congratulations! Your manuscript is now being handed over to our production team.

Kind regards,

on behalf of

Dr Samuel Kofi Tchum

Academic Editor

PLOS ONE